# The Clinical Outcome of Early Periprosthetic Joint Infections Caused by *Staphylococcus epidermidis* and Managed by Surgical Debridement in an Era of Increasing Resistance

**DOI:** 10.3390/antibiotics12010040

**Published:** 2022-12-27

**Authors:** Nada S. Shabana, Gesine Seeber, Alex Soriano, Paul C. Jutte, Silvia Westermann, Glenn Mithoe, Loredana Pirii, Théke Siebers, Bas ten Have, Wierd Zijlstra, Djordje Lazovic, Marjan Wouthuyzen-Bakker

**Affiliations:** 1Department of Medical Microbiology and Infection Prevention, University Medical Center Groningen, University of Groningen, 9712 CP Groningen, The Netherlands; 2Department of Orthopaedic Surgery, Universitätsklinik für Orthopädie und Unfallchirurgie, Pius-Hospital Oldenburg, 26121 Oldenburg, Germany; 3Department of Infectious Diseases, Hospital Clinic of Barcelona, University of Barcelona, 08007 Barcelona, Spain; 4Department of Orthopaedic Surgery, University Medical Center Groningen, University of Groningen, 9713 GZ Groningen, The Netherlands; 5Certe, Department of Medical Microbiology, University of Groningen, 9700 RB Groningen, The Netherlands; 6Izore, Centre for Infectious Diseases Friesland, 8917 EN Leeuwarden, The Netherlands; 7Department of Orthopaedic Surgery, Martini Hospital, 9728 NT Groningen, The Netherlands; 8Department of Orthopaedic Surgery, Medical Center Leeuwarden, 8934 AD Leeuwarden, The Netherlands

**Keywords:** periprosthetic joint infection, *S. epidermidis*, resistance, surgical debridement

## Abstract

Introduction: A risk factor for the failure of surgical debridement in patients with early periprosthetic joint infections (PJI) is the presence of multidrug-resistant microorganisms. Staphylococcus epidermidis is one of the most isolated microorganisms in PJI and is associated with emerging resistance patterns. We aimed to assess the antibiotic resistance patterns of *S. epidermidis* in early PJIs treated with surgical debridement and correlate them to clinical outcomes. Material and Methods: A retrospective multicentre observational study was conducted to evaluate patients with an early PJI (within 3 months after the index arthroplasty) by *S. epidermidis* with at least two positive intraoperative cultures. Clinical failure was defined as the need for additional surgical intervention or antibiotic suppressive therapy to control the infection. Results: A total of 157 patients were included. The highest rate of resistance was observed for methicillin in 82% and ciprofloxacin in 65% of the cases. Both were associated with a higher rate of clinical failure (41.2% vs. 12.5% (p 0.048) and 47.3% vs. 14.3% (p 0.015)), respectively. Furthermore, 70% of the cases had reduced susceptibility to vancomycin (MIC ≥ 2), which showed a trend towards a higher failure rate (39.6% vs. 19.0%, NS). Only 7% of the cases were rifampin-resistant. Only the resistance to fluoroquinolones was an independent risk factor for clinical failure in the multivariate analysis (OR 5.45, 95% CI 1.67–17.83). Conclusion: *S. epidermidis* PJIs show a high rate of resistance. Resistance to fluoroquinolones is associated with clinical failure. Alternative prophylactic antibiotic regimens and optimising treatment strategies are needed to improve clinical outcomes.

## 1. Introduction

Total joint replacement is considered a very effective medical intervention, providing pain relief, restoring joint function, and improving the quality of life of patients [1,2]. Unfortunately, complications may occur in a subset of patients [3]. One of the most devastating complications is the development of a periprosthetic joint infection (PJI). A PJI remains one of the most challenging complications, as it leads to a prolonged hospital stay, multiple surgeries, functional incapacitation, and increased mortality [4]. PJIs developing in the early postsurgical period are treated with surgical debridement, antibiotics, and retention of the implant (DAIR). Despite this intervention, failure of a DAIR procedure still occurs in around 40% of patients [5]. One of the factors contributing to a high failure rate is multidrug resistance, which makes the infection more difficult to eradicate [6,7,8,9,10].

*S. epidermidis* is a pathogen frequently isolated in early PJIs, particularly in polymicrobial infections [11,12]. In recent years, the emergence of multidrug-resistant nosocomial lineages of *S. epidermidis* has been observed [13]. These lineages contain certain mutations causing rifampin resistance but also reduced susceptibility to vancomycin and teicoplanin, antibiotics that are routinely used in PJI treatment [14,15]. In addition, a high rate of resistant *S. epidermidis* strains to methicillin and fluoroquinolones have been described [16]. The clinical meaning of these observations on the treatment outcome of PJI caused by *S. epidermidis* is not clear.

The aim of this study was to determine the antibiotic resistance patterns of *S. epidermidis* strains in a large, multicentre cohort of early PJIs treated with DAIR and evaluate its clinical outcome according to resistance profiles.

## 2. Material and Methods

### 2.1. Study Design

We performed a retrospective, multicentre, observational study. The data of patients with an early postsurgical PJI diagnosed and treated with DAIR from January 2007 to December 2017 were evaluated from an existing database. This database contained the data collected from five hospitals in three different countries. Three of the hospitals are located in The Netherlands (University Medical Center Groningen (UMCG), Martini Hospital, and Medical Center Leeuwarden (MCL)), one in Spain (Hospital Clínic Barcelona), and one in Germany (Universitätsklinik für Orthopädie und Unfallchirurgie Pius-Hospital Oldenburg).

### 2.2. Studied Population

Patients were included if they were (i) diagnosed with an early PJI of any joint according to the 2018 International Consensus Meeting criteria [17], (ii) treated with DAIR, and (iii) had at least two intraoperative cultures positive with *S. epidermidis.* An early postsurgical infection was defined as those patients diagnosed with a PJI within three months after the index surgery [18]. Patients younger than 18 years old and those with less than two years of follow-up were excluded from the analysis unless they failed within that time period.

In general, all the patients were treated with intravenous (IV) antibiotics for 1 or 2 weeks before switching to an oral antibiotic treatment regimen for an additional 10 to 11 weeks (total treatment duration of 3 months). All the patients treated with rifampin were treated during the entire oral treatment period.

The main outcome parameters were clinical and microbiological failures. Clinical failure was defined as the clinical need for an additional DAIR to control the infection, prosthesis removal during follow-up, or the need for lifelong suppressive therapy due to persistent clinical signs of infection. Microbiological failure was defined as a relapse of infection with the same microorganism as the initial infection according to the antibiotic susceptibility pattern.

### 2.3. Data Collection

Multiple variables on patient characteristics, microbiology, surgical procedure, antibiotic treatment, and follow-up were evaluated. The susceptibility of the cultured strains for different antibiotics was measured via an automatic system (VITEK). Breakpoints as defined by EUCAST were used to classify strains as susceptible or resistant. In this regard, in the case of vancomycin, a breakpoint of ≤4 was classified as susceptible. A MIC of ≥2 mg/L but ≤4 mg/L for vancomycin was considered reduced susceptibility [13].

### 2.4. Statistical Analysis

Continuous variables were summarised as means and standard deviations (SD) or as medians and interquartile ranges (IQRs) depending on data normality. Categorical variables were presented in percentages and absolute frequencies. A chi-square test was used to compare the categorical variables between groups, and Student’s *t*-test was used for the continuous variables if data were normally distributed. Logistic regression analysis was performed to identify the independent risk factors associated with clinical or microbiological failure. Variables with a *p* value ≤ 0.2 in the univariate analysis were included in the multivariate regression analysis. The analysis was performed using the Statistical Package for Social Science (SPSS), version 27.0 (IBM, Armonk, NY, USA). A *p* value < 0.05 was considered statistically significant.

## 3. Results

### 3.1. Studied Population

A total of 157 patients met the inclusion criteria and were included in this study. Appendix A shows the flowchart of the inclusions. Most of the included joints were hips (112, 71.3%), followed by knees (43, 27.4%), and shoulders (2, 1.3%). Osteoarthritis (OA) was the most common reason for primary prosthesis implantation (84.1%), followed by fracture (14%) and avascular necrosis (1.9%). Most of the cases had a polymicrobial infection (56.7%). The mean age of the study population was 73 years old (±9.69 SD), and 93 (59.2%) were females. The mean body mass index (BMI) was 31 kg/m^2^ (±6.60 SD). Almost half of the studied population had an ASA score of 3 (48%).

### 3.2. Resistance Profiles

The resistance patterns of *S. epidermidis* isolates are depicted in Figure 1. The highest resistance rate was observed for methicillin in 82% of the cases, followed by ciprofloxacin (65%), levofloxacin (59%), clindamycin (57%), cotrimoxazole (51%), and doxycycline (38%). The lowest rate of resistance was found for rifampin (7%). All the isolates were susceptible to vancomycin. In 70% of the cases, a MIC of ≥2 mg/L was measured using VITEK, indicating a reduced susceptibility to vancomycin. No additional E-tests were performed to determine the exact MIC. We did not find an association between a MIC of ≥2 mg/L for vancomycin and co-resistance with rifampin: Four rifampin-resistant strains had a vancomycin MIC ≥ 2 mg/L, while the other two rifampin-resistant strains had a vancomycin MIC < 2 mg/L (odds ratio = 0.553, *p*-value = 0.602). We additionally tested for co-resistance. Notably, 64% of the methicillin-resistant strains were resistant to clindamycin, 42% to doxycycline, and 70% to fluoroquinolones. In fluoroquinolone-resistant strains, 64% of the strains were resistant to cotrimoxazole, 80% to clindamycin, and 46% to doxycycline. With the exclusion of methicillin resistance, 6% of the tested strains were fully susceptible to all the tested antibiotics.

### 3.3. Clinical Outcomes According to Resistance Profiles and Oral Antimicrobial Regimen

Clinical failure was observed in 64 out of the 157 cases (40.8%) and microbiological failure with *S. epidermidis* in 7 out of 157 (4.5%). The majority of the clinically failed cases failed during targeted antibiotic treatment (92%). Of the 64 patients, 39 failed because they needed a second DAIR procedure to control the infection (61%), and 22 out of the 64 failed cases (34%) occurred because the implant needed to be removed. Furthermore, 59% of the patients had positive cultures during the second DAIR (23 out of 39) and 50% during implant removal (11 out of 22). Only seven cases had positive cultures with *S. epidermidis* during the second DAIR (and were classified as microbiological failure). Table 1 shows the results of the uni- and multivariate analysis. Fluoroquinolone resistance was the only independent predictor for treatment failure in the initial analysis, but due to large confidence intervals, we tested for collinearity. We observed collinearity between fluoroquinolone resistance and resistance to methicillin, cotrimoxazole, and clindamycin. Therefore, these variables were excluded from the multivariate analysis. Resistance to fluoroquinolones remained the only independent predictor for clinical failure (OR 5.45, 95% CI 1.67–17.83).

A higher rate of clinical failure was observed in cases with a vancomycin MIC ≥ 2 mg/L than cases with a vancomycin MIC < 2 mg/L (39.6% vs. 19.0%), but this difference was not statistically significant. However, all seven microbiological failures had a MIC ≥ 2 mg/L to vancomycin, and five of them were treated with vancomycin IV before switching to an oral regimen. All five patients had serum levels of vancomycin (ranging between 20 and 25 mg/L). Only one of the seven patients was treated with a rifampin-based antibiotic regimen.

Most patients, i.e., 55 out of the 157 cases (35%), were treated with an oral regimen containing a fluoroquinolone, followed by clindamycin (21%), linezolid (14.6%), tetracycline (12.7%), and cotrimoxazole (6.4%). Among the 157 patients, 96 received a rifampin-based regimen (61.1%).

Patients with fluoroquinolone resistance were treated with the following oral antibiotic therapy: A rifampin-based regimen was administered in 69.1%, from which 33.3% used cotrimoxazole, 33.3% used clindamycin, 29.4% used linezolid, and 26.7% used tetracycline as co-antibiotic. Of the patients treated with monotherapy, 32.5% used tetracycline, 30.6% used cotrimoxazole, 30.2% used clindamycin, and 29.7% used linezolid.

None of the used oral antibiotic regimens was superior over the other: Clinical failure using cotrimoxazole was 40.0% versus 40.8% for the other oral regimens (*p* 0.96); for clindamycin, this rate was 51.5% vs. 37.9% for the other regimens (*p* 0.16); tetracycline had a failure rate of 45.0% vs. 40.1% for the other regimens (*p* 0.68); for fluoroquinolone, it was 38.2% vs. 42.2% for the other regimens (*p* 0.63); in linezolid, clinical failure was 39.1% vs. 41.0% for the other regimens (*p* 0.86); the failure rate of the rifampin-based regimen was 38.5% vs. 44.3% for non-rifampin based regimens (*p* 0.48). The failure rate of a rifampin-based regimen using fluoroquinolone as the antibiotic backbone was 34.1% vs. 43.4% for the other antibiotic regimens (*p* 0.29). The number of microbiological failures was too small to perform statistical analyses.

## 4. Discussion

This study aimed to assess the clinical outcome of patients with an early acute PJI caused by *S. epidermidis* and treated with surgical debridement and to determine whether certain resistance profiles are associated with clinical or microbiological failure. In our cohort of 157 patients, we found a relatively high rate of clinical failure (41%) but relatively low microbiological failure due to *S. epidermidis* (4.5%). The majority of cases failed during targeted antibiotic treatment. The highest rate of resistance in our cohort was found for methicillin, followed by fluoroquinolones and clindamycin. The presence of fluoroquinolone resistance was an independent predictor of clinical failure. Remarkably, no association was found between failure and the type of targeted antibiotic treatment.

The *S. epidermidis* resistance profiles we observed in our cohort are in accordance with previous studies. A study published by Zeller et al. reported on the epidemiology profile of bone and joint infections from a large French referral centre over the last 12 years. They showed that the *S. epidermidis* strains isolated from these infections had a methicillin resistance of merely 84% (17). A study evaluating the outcome of PJI patients with methicillin-sensitive and methicillin-resistant *S. epidermidis* strains demonstrated a lower infection eradication rate in the presence of methicillin resistance (95.2% vs. 54.2%) in patients with a chronic PJI treated with a two-stage exchange procedure [19]. This association was not found in a large cohort of methicillin-susceptible and -resistant acute *S. aureus* PJIs treated with surgical debridement [20]. We did find a lower rate of clinical success in the presence of methicillin resistance in early PJIs treated with surgical debridement; however, only the resistance to fluoroquinolones was an independent predictor of failure in the multivariate analysis. Fluoroquinolones are considered as the first first-line treatment for staphylococcal PJIs when combined with rifampin [14,21]. Remarkably, we did not find an association between clinical or microbiological failure and targeted antibiotic treatment. Most of the failures occurred while still being under antibiotic treatment. These cases required additional surgery (either a second debridement or implant removal), and around half of them still had positive cultures during surgery but rarely due to persistent *S. epidermidis*. This finding suggests that resistant strains are a surrogate marker for failure rather than the need for treatment with alternative, second-line antibiotic regimens. This finding is in line with a previous publication where antibiotic resistance itself, and not the type of antibiotic treatment, predicted failure [6]. In addition, the majority of the failures occurred while the patient was under antibiotic treatment, targeted towards the initially isolated microorganisms. For this reason, in culture-negative failures, the persistence of infection with *S. epidermidis* cannot be ruled out. Clinical failure in this regard might be due to suboptimal antibiotic treatment or suboptimal surgical debridement. As for culture-positive failures in patients who initially had a monomicrobial infection with *S. epidermidis*, reinfection occurred with another microorganism that was probably introduced during the first DAIR procedure. Unfortunately, we did not investigate whether the patients received antimicrobial prophylaxis prior to the DAIR procedure.

There have been contradictory results as to whether the reduced susceptibility of staphylococci to vancomycin is associated with a worse outcome. A meta-analysis performed by Ishaq et al. evaluating patients with methicillin-resistant *S. aureus* bacteraemia did not find a significant association between mortality or persistent bacteraemia and a higher vancomycin MIC in the susceptibility range (i.e., MIC ≥ 1.5 mg/L [22]. One of the possible explanations the authors propose for the lack of a significant association is the inclusion of those studies that include heteroresistant vancomycin-intermediate *S. aureus* (hVISA), a strain that is known to be less virulent. Some studies do show a clear association between high vancomycin MICs and infection recurrence [22,23]. In our cohort, a higher clinical failure rate was observed in cases with a vancomycin MIC ≥ 2.0 mg/L, but this finding was not statistically significant. A limitation of our analysis was that the MIC levels for vancomycin were only determined using VITEK, and no additional E-tests were performed. Based on the findings reported in the literature, we expected an association between reduced susceptibility to vancomycin and co-resistance with rifampin [13]. Although we did observe a reduced susceptibility to vancomycin in 70% of the cases, rifampin resistance in our cohort was low (7%). We did not evaluate whether the patients in our cohort were previously treated with vancomycin combined with rifampin, as this combination pre-exposes patients to rifampin resistance [24,25]. In addition, microbiological failure due to *S. epidermidis* in our cohort was low (<5%), hampering the ability to evaluate the emergence of resistance under antibiotic therapy.

Aside from the limitations already mentioned, a selection bias may have occurred during the study period, as some coagulase-negative staphylococci that were cultured were not typed to determine their species. For this reason, we could have missed some PJIs caused by *S. epidermidis*, as these cases could not be included. Second, it is hard to determine the causality between resistance patterns and failure, as many factors (including host factors) predispose patients to clinical failure.

In conclusion, patients with an early PJI caused by *S. epidermidis* and treated with surgical debridement have a relatively high rate of clinical failure and depict a high resistance rate to first-line prophylactic and antimicrobial treatment agents. Considering the high rate of methicillin resistance, our findings stress the importance of optimising antimicrobial prophylaxis in the primary procedure. In addition, the high rate of reinfections occurring after the first DAIR procedure advocates the need for antimicrobial prophylaxis during a DAIR procedure. Patients with an early *S. epidermidis* PJI with resistance to fluoroquinolones have a higher rate of clinical failure, but its underlying mechanism still needs to be elucidated.

## Figures and Tables

**Figure 1 antibiotics-12-00040-f001:**
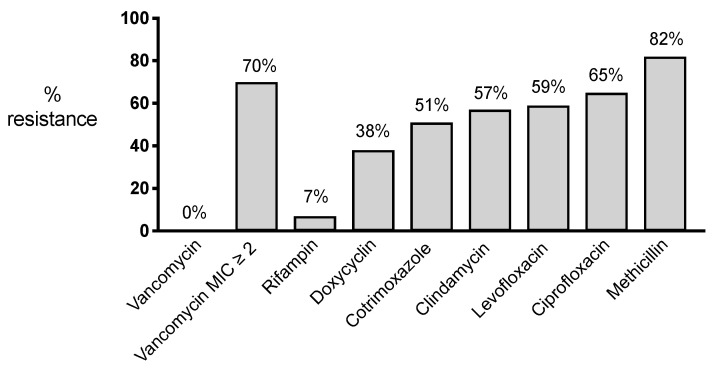
Percentage of resistance for different antibiotics in *S. epidermidis* strains isolated from patients with an early PJI (*n* = 157).

**Table 1 antibiotics-12-00040-t001:** Risk factors for clinical failure in patients with early PJI caused by *S. epidermidis*.

	Univariate Binary Regression Analysis			Multivariate Binary Regression Analysis	
Variable	Non-Failures (*n* = 93)	Failures (*n* = 64)	*p* Value	Adjusted OR (95% CI)	*p* Value
**Baseline characteristics** Male sex	37.6%	45.3%	0.34		
Age > 70	66.7%	82.8%	0.03 *	0.97 (0.24–2.05)	0.97
BMI > 30	59.1%	70.3%	0.15 *	0.71 (0.25–3.90)	0.53
ASA (≥3)	44.1%	54.7%	0.19 *	1.03 (0.35–3.05)	0.96
**Comorbidities** Arterial hypertension	67.7%	67.2%	0.94		
Heart failure	9.7%	9.4%	0.95		
Coronary heart disease	15.1%	20.3%	0.39		
Diabetes mellitus	20.4%	23.4%	0.65		
Chronic renal failure	7.5%	17.2%	0.06 *	1.48 (0.31–7.22)	0.63
COPD	19.4%	14.1%	0.39		
Liver cirrhosis	2.2%	1.6%	0.79		
Rheumatoid arthritis	7.5%	4.7%	0.47		
**Characteristics implant** Hip	72.0%	70.3%	0.81		
Fracture Cemented	10.8% 74.2%	18.8% 82.8%	0.16 * 0.20 *	2.20 (0.59–8.23) 3.23 (0.49–21.39)	0.24 0.23
**Microorganism** Polymicrobial	61.2%	62.7%	0.85		
**Resistance profile** Methicillin resistance Fluoroquinolone resistance Cotrimoxazole resistance Clindamycin resistance Doxycycline resistance Rifampin resistance Vancomycin MIC ≥ 2	76.3% 49.2% 42.1% 49.2% 39.0% 3.4% 64.4%	93.3% 83.9% 66.7% 73.3% 37.5% 12.9% 70.8%	0.05 ** 0.001 * 0.03 ** 0.03 ** 0.90 0.09 * 0.59	5.45 (1.67–17.83) 5.69 (0.61–52.75)	0.005 0.13

* Variables with a *p* value ≤ 0.2 were included in the multivariate binary regression analysis. ** Due to the collinearity between fluoroquinolone resistance and resistance to methicillin, cotrimoxazole, and clindamycin, these variables were excluded from the multivariate analysis. OR, odds ratio; CI, confidence interval; BMI, body mass index; ASA; American Society of Anaesthesiologists; COPD, chronic obstructive pulmonary disease.

## Data Availability

Not applicable.

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
