# Peer review of "The Clinical Outcome of Early Periprosthetic Joint Infections Caused by Staphylococcus epidermidis and Managed by Surgical Debridement in an Era of Increasing Resistance"

_antibiotics, 2022, doi:10.3390/antibiotics12010040_

Round 1

Reviewer 1 Report

Dear authors,

First, I would like to express sincere gratitude to get the opportunity to review your manuscript. The effort of the author is appreciated. Congratulations on the subject of the manuscript. We are in an era of increasing resistance. A well-structured manuscript. 

After assessing the manuscript, the following issues raised my concerns or represent suggestions that in my opinion could increase the quality of the manuscript: 

-       Please reassess the manuscript for formatting and typing mistakes.

-    Please correct the mistake in the name of one of the authors, “Loredana Piiri”, it should be “Loredana Pirii”

-      Please add the missing data for affiliations no.5 and 6

-  In the “studied population” you reported the following “Most of the included joints were hips 112 (71.3%), followed by knees 43 (27.4%), and shoulders (1.3%).” As a suggestion, report also the no. not just the % for shoulder PJIs. The same suggestion for the next sentence where you report the no just for AVN.

The weakness I felt is that the sample numbers can be accumulated more to solidify your conclusion, this disadvantage is overcome by the fact that it is a multicenter study,  nevertheless, the results definitely should be taken into consideration.

Author Response

Reviewer 1

First, I would like to express sincere gratitude to get the opportunity to review your manuscript. The effort of the author is appreciated. Congratulations on the subject of the manuscript. We are in an era of increasing resistance. A well-structured manuscript. 

After assessing the manuscript, the following issues raised my concerns or represent suggestions that in my opinion could increase the quality of the manuscript: 

-       Please reassess the manuscript for formatting and typing mistakes.

Response: we reassessed and made some necessary changes.

-    Please correct the mistake in the name of one of the authors, “Loredana Piiri”, it should be “Loredana Pirii”

Response: adapted

-      Please add the missing data for affiliations no.5 and 6

Response: adapted

-  In the “studied population” you reported the following “Most of the included joints were hips 112 (71.3%), followed by knees 43 (27.4%), and shoulders (1.3%).” As a suggestion, report also the no. not just the % for shoulder PJIs. The same suggestion for the next sentence where you report the no just for AVN.

Response: thank you for noticing these small errors, we have added the absolute numbers on shoulders and removed it for AVN

The weakness I felt is that the sample numbers can be accumulated more to solidify your conclusion, this disadvantage is overcome by the fact that it is a multicenter study,  nevertheless, the results definitely should be taken into consideration.

Reviewer 2 Report

The article discusses the antibiotic resistance of S. Epidermidis in prosthesis infections. The population considered had undergone prosthetic replacement surgery within the previous three months.

- The introduction is well done, concise but quite comprehensive. 

-  Results are quite clear. However, it would be desirable to know whether cross-resistance exists and the treatment applied after the discovery of resistance. It would also be helpful in how many of these cases developed septic states, the level of some markers as Procalcitonin, and how they were diagnosed and treated. In this regard, I suggest two articles that might help achieve this goal:

1.  Postmortem diagnosis of sepsis: A preliminary immunohistochemical study with an anti-procalcitonin antibody.

2. Expression of MicroRNAs in Sepsis-Related Organ Dysfunction: A Systematic Review.

- Nothing to add about discussion paragraph;

- I would add a "conclusion paragraph" in which I would put everything written after line 232 and add some comments regarding the need for prevention of bacterial resistance.

In line 74 please replace ";" with ":"

Author Response

Reviewer 2

The article discusses the antibiotic resistance of S. Epidermidis in prosthesis infections. The population considered had undergone prosthetic replacement surgery within the previous three months.

- The introduction is well done, concise but quite comprehensive. 

Response: thank you

-  Results are quite clear. However, it would be desirable to know whether cross-resistance exists and the treatment applied after the discovery of resistance. It would also be helpful in how many of these cases developed septic states, the level of some markers as Procalcitonin, and how they were diagnosed and treated. In this regard, I suggest two articles that might help achieve this goal:

  1. Postmortem diagnosis of sepsis: A preliminary immunohistochemical study with an anti-procalcitonin antibody.
  2. Expression of MicroRNAs in Sepsis-Related Organ Dysfunction: A Systematic Review

Response: We have added some numbers on cross-resistance and the antibiotic treatment. None of the patients became septic. The use of procalcitonin is beyond the scope of our paper and was not performed.

- Nothing to add about discussion paragraph;

- I would add a "conclusion paragraph" in which I would put everything written after line 232 and add some comments regarding the need for prevention of bacterial resistance.

Response: thank you for the suggestion, we have expanded the conclusion paragraph in the revised version.

In line 74 please replace ";" with ":"

Response: adapted

Reviewer 3 Report

I read with interest the manuscript “The Clinical Outcome of Early Periprosthetic Joint Infection Caused by Staphylococcus Epidermidis in an era of Increasing Resistance” by Shabhana NS and collaborators. It contains original information on the topic and is, in general terms, well written and easy to follow. However, at its present form, I believe it should not be accepted for publication, unless several points are considered for review.

- The manuscript contains information on the resistance profiles of S. epidermidis causing early PJI and managed with DAIR and links these resistance data with the outcome of patients. However, from my point of view, the data is merely descriptive and only simple comparisons have been performed, without adjusting for potential confounders. Indeed, a greater deepening on the data should be considered, especially regarding the outcome analysis. Reading the manuscript, I imagine that the authors possess data on failure dates which may prompt them to perform a survival analysis and evaluate factors associated with the outcome by multivariate Cox regression. Similarly, a Kaplan Meier analysis may be worth considering. Actually, the authors refer in the Methods section that logistic regression methodology has been employed for univariate and multivariate analysis, but this is not included in the manuscript (only a univariate OR is included, when referring to the impact of vancomycin MIC). The same authors acknowledge their limitation on the discussion section, but I do not understand then, why they have not performed such analyses.

- Similarly, when exploring the outcomes according to resistance patterns, it is vital to understand how patients have been treated. The information included by the authors in the last paragraph of the last subsection of Results refers to the whole cohort. However, in order to understand, how methicillin susceptibility, vancomycin resistance or others impact, it is important to understand how these particular patients have been treated in terms of initial iv treatment (and how long) and subsequent oral treatment (and how long). Moreover, the role of rifampin is superficially referred to; was a survival bias considered when evaluating its role? Were patients categorized as having received rifampin independently of the duration of rifampin treatment?

- The authors consider two different outcomes, clinical failure and microbiological failure. Clinical failure included the need for additional DAIR, prosthesis removal or SAT. Were any time cutoffs used to define this clinical failure? For example, a patient could be considered failure if prosthesis was removed beyond 2 years since DAIR; in such a case, I am not sure whether the patient should be considered as failed, especially if the reason was orthopedic. Were intraoperative cultures taken into account when subsequent DAIRs or prosthesis removal were performed? Microbiological failure includes only those with positive cultures, but what about patients who received a second DAIR within a short time period since the first, with negative intraoperative cultures and were receiving appropriate antimicrobial treatment (when the second DAIR was performed)? Actually, there is a high discrepancy between the rates of clinical failure (very high) and microbiological failure (very low). Providing more data on causes of failure and time of failure would be useful to understand what happened with the cohort and whether such failures can be fully attributed to the surgical procedures or the antimicrobial treatment.

Other points:

- Consider modifying the article title to “ The Clinical Outcome… by S. epidermidis managed by DAIR in an era…”.

- Please review affiliations 5 and 6.

- Line 34 and 35 “Fluoroquinolones are, when combined with rifampin, the first line…”. Although I agree with the sentence, how is this sentence supported by the results of the abstract? O even from results of the manuscript? According to your results, clinical outcome is similar in those receiving FQ or not and those receiving rifampin or not… Please clarify.

- Line 50. Emergence.

- Line 52. Reduced.

- Line 55. Please modify to “… treatment outcome of PJI by S. epidermidis…”

- Line 76. Some consider that early infections are those within 1 month since index surgery, although DAIR may be extended to those with infections in the first 3 months after surgery. Please support your definition with a reference or clarify according to current PJI classifications.

- Line 116. More than half of episodes were polymicrobial. What other microorganisms were involved? Were results/outcome similar between monomicrobial or polymicrobial? I imagine that the impact of S. epidermidis may be different in mono vs polymicrobial, as in the latter the inoculum of S. epidermidis is mixed with other microorganisms, which may be more responsible for failure? What is your opinion?

- Line 126. No additional e-tests were performed but this is not in accordance to what is stated in Methods section. Please clarify.

- Lines 149-150. Methicillin resistance was associated with higher failure rates than susceptibility. Although it is a different microorganism, this is in discordance with a large study of PJI by S. aureus managed with DAIR, where MSSA and MRSA episodes had similar outcomes (Lora-Tamayo et al. Clin Infect Dis 2013;56(2):182–94). This may be related with the definitions on the outcomes. May be worth a discussion.

- Lines 155-158. Vancomycin was mostly used in those patients with microbiological failure, who had high MICs. Serums were considered to be appropriate when 20-25 mg/L. This is a controversial issue as there is ongoing debate whether trough concentrations or AUC/MIC should be used as PK/PD target correlating with efficacy. If AUC/MIC is chosen, with MIC >2 mg/L, dosing may be insufficient and other antimicrobial alternatives might be worth considering. Also, most studies on vancomycin monitoring refer to MRSA and not S. epidermidis. Finally, what about the correlation between plasma and site concentrations? 

- In this line, would daptomyicin be a better option for these patients? There are preclinical data with S. aureus suggesting that dapto may have anti-biofilm activity and have high efficacy in combination with rifampin. This may be discussed in the manuscript.

- Line 166. Clinical failure was higher with clindamycin, although non-significant. This is in discrepancy with data recently published by your group (Beldman et al. Clin Infect Dis. 2021 Nov 2;73(9):1634-1641.); the total population is a bit different, but in this article clindamycin had good efficacy. Please discuss.

- Line 176. Asterisks in Figure 2 have not been defined.

- Line 210. Reference 20 is not an article by Ishaq et al. Actually, there are two reference 19.

- Line 212. S. aureus, in italics.

- Finally, the inclusion of Tables to support your data may be worth considering.

Author Response

Reviewer 3

I read with interest the manuscript “The Clinical Outcome of Early Periprosthetic Joint Infection Caused by Staphylococcus Epidermidis in an era of Increasing Resistance” by Shabhana NS and collaborators. It contains original information on the topic and is, in general terms, well written and easy to follow. However, at its present form, I believe it should not be accepted for publication, unless several points are considered for review.

- The manuscript contains information on the resistance profiles of S. epidermidis causing early PJI and managed with DAIR and links these resistance data with the outcome of patients. However, from my point of view, the data is merely descriptive and only simple comparisons have been performed, without adjusting for potential confounders. Indeed, a greater deepening on the data should be considered, especially regarding the outcome analysis. Reading the manuscript, I imagine that the authors possess data on failure dates which may prompt them to perform a survival analysis and evaluate factors associated with the outcome by multivariate Cox regression. Similarly, a Kaplan Meier analysis may be worth considering. Actually, the authors refer in the Methods section that logistic regression methodology has been employed for univariate and multivariate analysis, but this is not included in the manuscript (only a univariate OR is included, when referring to the impact of vancomycin MIC). The same authors acknowledge their limitation on the discussion section, but I do not understand then, why they have not performed such analyses.

Response: we completely agree, we have added the multivariate analysis to the revised version of the paper.

- Similarly, when exploring the outcomes according to resistance patterns, it is vital to understand how patients have been treated. The information included by the authors in the last paragraph of the last subsection of Results refers to the whole cohort. However, in order to understand, how methicillin susceptibility, vancomycin resistance or others impact, it is important to understand how these particular patients have been treated in terms of initial iv treatment (and how long) and subsequent oral treatment (and how long). Moreover, the role of rifampin is superficially referred to; was a survival bias considered when evaluating its role? Were patients categorized as having received rifampin independently of the duration of rifampin treatment?

Response: We understand its importance. However, it is difficult in terms of cross resistance how to display these details. In general, all patients have been treated with 1 or 2 weeks of IV treatment and an additional 10 to 11 weeks of oral treatment (total treatment duration of 3 months). All patients treated with rifampin were treated during the entire oral treatment period. We have added this information to the material and method section of the paper to give a clearer picture.

- The authors consider two different outcomes, clinical failure and microbiological failure. Clinical failure included the need for additional DAIR, prosthesis removal or SAT. Were any time cutoffs used to define this clinical failure? For example, a patient could be considered failure if prosthesis was removed beyond 2 years since DAIR; in such a case, I am not sure whether the patient should be considered as failed, especially if the reason was orthopedic. Were intraoperative cultures taken into account when subsequent DAIRs or prosthesis removal were performed? Microbiological failure includes only those with positive cultures, but what about patients who received a second DAIR within a short time period since the first, with negative intraoperative cultures and were receiving appropriate antimicrobial treatment (when the second DAIR was performed)? Actually, there is a high discrepancy between the rates of clinical failure (very high) and microbiological failure (very low). Providing more data on causes of failure and time of failure would be useful to understand what happened with the cohort and whether such failures can be fully attributed to the surgical procedures or the antimicrobial treatment.

Response: Thank you for bringing up this important aspect. We did not used any time cut-offs for clinical failure, especially since S. epidermidis can cause low-grade infection that may result in failure many years after the initial infection. We agree with the reviewer that the requested information is important to add. The majority of cases failed while still being under antibiotic treatment (92%). 39 out of 64 patients failed because they needed a second DAIR procedure to control the infection (61%). In 22 out of the 64 failed cases (34%) the implant needed to be removed (14% of the total cohort). 59% of patients had positive cultures during the second DAIR (23 out of 39) and 50% during implant removal (11 out of 22). Only 7 cases still had positive cultures with S. epidermidis during the second DAIR (and were classified as microbiological failure). Since we could not find an association between the antibiotic treatment itself and clinical failure, and the microbiological failure is low, antibiotic resistance might be a surrogate marker for failure. This is also in line with other literature. We have added more detailed information on failure in the result section of the paper and added a subsection in the discussion section of the manuscript discussing this important point.

Other points:

- Consider modifying the article title to “ The Clinical Outcome… by S. epidermidis managed by DAIR in an era…”.

Response: adapted

- Please review affiliations 5 and 6.

Response: adapted

- Line 34 and 35 “Fluoroquinolones are, when combined with rifampin, the first line…”. Although I agree with the sentence, how is this sentence supported by the results of the abstract? O even from results of the manuscript? According to your results, clinical outcome is similar in those receiving FQ or not and those receiving rifampin or not… Please clarify.

Response: We agree, also in line with the previous comment of the reviewer, we have adapted this part.

- Line 50. Emergence.

Response: thank you, adapted

- Line 52. Reduced.

Response: thank you, adapted

- Line 55. Please modify to “… treatment outcome of PJI by S. epidermidis…”

Response: thank you, adapted

- Line 76. Some consider that early infections are those within 1 month since index surgery, although DAIR may be extended to those with infections in the first 3 months after surgery. Please support your definition with a reference or clarify according to current PJI classifications.

Response: Thank you, we have added a reference. In general, acute infections in the post-surgical period are those within 1 month, but early infections are those defined within 3 months.

- Line 116. More than half of episodes were polymicrobial. What other microorganisms were involved? Were results/outcome similar between monomicrobial or polymicrobial? I imagine that the impact of S. epidermidis may be different in mono vs polymicrobial, as in the latter the inoculum of S. epidermidis is mixed with other microorganisms, which may be more responsible for failure? What is your opinion?

Response: Thank you, we have added this in the multivariate analysis of the paper. Polymicrobial PJIs did not had a higher clinical rate of failure. Unfortunately, we do not have detailed information on the other microorganisms for the polymicrobial cases in this database.

- Line 126. No additional e-tests were performed but this is not in accordance to what is stated in Methods section. Please clarify.

Response: Agreed, we have adapted this.

- Lines 149-150. Methicillin resistance was associated with higher failure rates than susceptibility. Although it is a different microorganism, this is in discordance with a large study of PJI by S. aureus managed with DAIR, where MSSA and MRSA episodes had similar outcomes (Lora-Tamayo et al. Clin Infect Dis 2013;56(2):182–94). This may be related with the definitions on the outcomes. May be worth a discussion.

Response: Thank you for this suggestion. We have added a part to the discussion section of the paper. In fact, this finding is in accordance with our multivariate analysis, in which only resistance to fluoroquinoles were independently associated with a higher rate of clinical failure.

- Lines 155-158. Vancomycin was mostly used in those patients with microbiological failure, who had high MICs. Serums were considered to be appropriate when 20-25 mg/L. This is a controversial issue as there is ongoing debate whether trough concentrations or AUC/MIC should be used as PK/PD target correlating with efficacy. If AUC/MIC is chosen, with MIC >2 mg/L, dosing may be insufficient and other antimicrobial alternatives might be worth considering. Also, most studies on vancomycin monitoring refer to MRSA and not S. epidermidis. Finally, what about the correlation between plasma and site concentrations? 

Response: We agree that this is an interesting point to discuss. However, since we did not found a strong association between higher vancomycin MICs and failure (not statistically significant), and these failures were, in addition not treated with rifampin, we feel that discussing this point in too much detail goes beyond the scope of the paper. We did remove the word “adequate” concerning the serum levels of vancomycin.

- In this line, would daptomyicin be a better option for these patients? There are preclinical data with S. aureus suggesting that dapto may have anti-biofilm activity and have high efficacy in combination with rifampin. This may be discussed in the manuscript.

Response: See my previous response.

- Line 166. Clinical failure was higher with clindamycin, although non-significant. This is in discrepancy with data recently published by your group (Beldman et al. Clin Infect Dis. 2021 Nov 2;73(9):1634-1641.); the total population is a bit different, but in this article clindamycin had good efficacy. Please discuss.

Response: Since the difference was not statistically significant compared to other regimens, we feel that this out of the scope of the paper to discuss in detail, in particular since overall, the resistance pattern rather than the antibiotic treatment itself was associated with a higher failure rate.

- Line 176. Asterisks in Figure 2 have not been defined.

Response: thank you, adapted

- Line 210. Reference 20 is not an article by Ishaq et al. Actually, there are two reference 19.

Response: thank you, adapted

- Line 212. S. aureus, in italics.

Response: thank you, adapted

- Finally, the inclusion of Tables to support your data may be worth considering.

Response: we have added an additional Table to the revised version.

Reviewer 4 Report

In this interesting work the authors provide interesting information on the PJI produced by S. epidermidis and in general, the data shown support their conclusions. Please find below some comments:

Line 93. Could you explain what do you mean with “i.e. break point 4”? Did you use this breakpoint for all antibiotics?

Line 115. As most of the cases had a polymicrobial infection, please provide details about the other bacteria involved and their frequency.

Line 120. As for resistance patterns, how many S. epidermidis were resistant to 1,2,3 antibiotics?

Line 236.  Taking into account that biofilms (relevance, ability of S. epidermidis to form biofilms, etc.) are not mentioned in the entire manuscript, it is surprising that the final reflection is regarding the importance of improving treatment strategies against bacterial biofilms. The problem of biofilms should be addressed in the text; if not, the final conclusion has to be modified.

Author Response

Reviewer 4

In this interesting work the authors provide interesting information on the PJI produced by S. epidermidis and in general, the data shown support their conclusions. Please find below some comments:

Line 93. Could you explain what do you mean with “i.e. break point ≤4”? Did you use this breakpoint for all antibiotics?

Response: Only for vancomycin, we have clarified this now in the text.

Line 115. As most of the cases had a polymicrobial infection, please provide details about the other bacteria involved and their frequency.

Response: Unfortunately, we do not have detailed information on the other microorganisms in this database for the polymicrobial cases.

Line 120. As for resistance patterns, how many S. epidermidis were resistant to 1,2,3 antibiotics?

Response: We have added the data concerning cross resistance to the result section of the paper

Line 236.  Taking into account that biofilms (relevance, ability of S. epidermidis to form biofilms, etc.) are not mentioned in the entire manuscript, it is surprising that the final reflection is regarding the importance of improving treatment strategies against bacterial biofilms. The problem of biofilms should be addressed in the text; if not, the final conclusion has to be modified.

Response: Since the microbiological failure was really low in this study, we have chosen not to expand on other agents for the treatment of S epidermidis. Also since there was no correlation between antibiotic treatment and clinical failure. We have added a part to the discussion section of the paper.

Round 2

Reviewer 3 Report

The manuscript “The Clinical Outcome of Early Periprosthetic Joint Infection Caused by Staphylococcus Epidermidis and Managed By Surgical Debridement in an era of Increasing Resistance” has clearly improved after the first revision. The effort to meet the reviewers’ comments and suggestions should be acknowledged. However, my impression is that there are still some points that need revision, some of them having been mentioned in my previous review. 

- The new version of the manuscript provides a multivariate analysis to evaluate factors associated with clinical failure, which has been done by logistic regression. I understand, from the Methods section, that authors have data on follow-up and when events (failures) occurred. Being that so and having time-to-event data, I do not understand, then, why authors do not evaluate factors associated with the outcome by a Cox regression model, providing hazard ratios and confidence intervals. I suggest that authors do so in Table 1. I also suggest that a Kaplan Meier survival curve is showed, either in the main body or supplementary material, so journal readers can understand how and when clinical failure presents in the cohort.

- Table 1 shows 95% CI for OR and these are significantly wide. I would recommend authors to test in their dataset whether there might be collinearity between some covariates, particularly between those under resistance profile subsection. Most patients had cross resistance between antibiotic families, so it might be that some resistance covariates are measuring the same phenomenon. Perhaps only introducing one covariate (fluoroquinolone resistance?) in the multivariate analysis might solve the issue.

- I also suggest that actual counts (and not only percentages) are provided for each covariate in the univariate analysis in Table 1. 

- Regarding resistance profiles and outcomes, I believe there is a discrepancy between univariate p values shown in Table 1 and p values shown in Figure 2. I understand that p values should be the same, since Table 1 show column percentages (denominator being failure and non-failure and numerator being resistant isolates) and Figure 2 show row percentages (denominator being resistant and susceptible isolates and numerator being failure). Please review.

- If possible, it would be interesting, as mentioned in my previous review, to show how patients with susceptibility and resistance to antibiotics were treated. For example, what regimens received patients with infections by fluoroquinolone-susceptible and resistant isolates. Similarly, to other regimens. I believe this would enrich the content of the manuscript.

- Lines 238-241: According to your results, failure occurred mostly during targeted antibiotic treatment (92%), so I understand that in the first 3 months post-DAIR. The low microbiological failure, defined as positive cultures to the same S. epidermidis, should be understood in this context, since patients were receiving appropriate treatment. Superinfection was, then, frequent, since other patients had positive cultures too. The remaining had negative cultures, under appropriate treatment, but failed, being probably failures due to the same microorganism (persistence). Therefore, we do not know whether particular regimens would reduce the rate of persistent failures, apart from only the resistance profile. Moreover, other factors might play a role, such as the particular surgeon who performed the debridement or the type of hospital. Because of these reasons, I suggest that you reduce the strength of the sentences in these lines and turn them into potential reasonable hypotheses.
